# Development and Validation of the New Liquid Chromatography-Tandem Mass Spectrometry Method for the Determination of Unbound Tacrolimus in the Plasma Ultrafiltrate of Transplant Recipients

**DOI:** 10.3390/pharmaceutics14030632

**Published:** 2022-03-12

**Authors:** Magdalena Bodnar-Broniarczyk, Karola Warzyszyńska, Katarzyna Czerwińska, Dorota Marszałek, Natalia Dziewa, Maciej Kosieradzki, Tomasz Pawiński

**Affiliations:** 1Department of Drug Chemistry, Medical University of Warsaw, 02-097 Warsaw, Poland; dorota.marszalek@wum.edu.pl (D.M.); tomasz.pawinski@wum.edu.pl (T.P.); 2Department of General and Transplantation Surgery, Medical University of Warsaw, 02-014 Warsaw, Poland; natalia.dziewa@gmail.com (N.D.); mpkosieradzki@gmail.com (M.K.); 3Department of Transplantation Medicine and Nephrology, Medical University of Warsaw, 02-014 Warsaw, Poland; k.przychodzka@onet.eu

**Keywords:** tacrolimus, TDM, unbound concentration, ultrafiltration, validation, LC–MS/MS

## Abstract

(1) Background: Only unbound tacrolimus particles are considered to be active and capable of crossing cellular membranes. Thus, the free-drug concentration might be better associated with clinical effects than the total drug concentration used for dosage adjustment. We propose a new, fully validated online liquid chromatography-tandem mass spectrometry (LC-MS/MS) method for unbound tacrolimus concentration measurement. (2) Methods: The determination of the unbound tacrolimus concentration in plasma ultrafiltrate was performed with the Nexera LC system with LCMS-8050 triple quadrupole MS using ascomycin as an internal standard. Chromatographic separation was made using a HypurityC18 analytical column. MS/MS with electrospray ionization and positive-ion multiple-reaction monitoring was used. The unbound tacrolimus level was determined in 36 patients after solid organ transplantation (*n* = 140). (3) Results: A lower limit of quantification 0.1 pg/mL was achieved, and the assay was linear between 0.1 and 20 pg/mL (R^2^ = 0.991). No carry-over was detected. The within-run and between-run accuracies ranged between 97.8–109.7% and 98.3–107.1%, while the greatest imprecision was 10.6% and 10.7%, respectively. Free tacrolimus in patients’ plasma ultrafiltrate varied between 0.06 and 18.25 pg/mL (median: 0.98 pg/mL). (4) Conclusions: The proposed method can be easily implemented. The significance of the unbound tacrolimus concentration needs to be investigated. This may facilitate the individualization and optimization of immunosuppressive treatment.

## 1. Introduction

Tacrolimus (TAC) is the first-choice immunosuppressive agent after solid organ transplantation [1,2]. Although the safety and efficacy of TAC are well established, drug doses need to be targeted in a narrow therapeutic window. Underexposure may result in acute rejection of the transplanted organ, whereas too high TAC concentrations may lead to TAC toxicity [3,4,5]. Both increase morbidity and mortality rates in solid organ recipients [1,2,6,7]. Thus, TAC-based treatment requires careful therapeutic drug monitoring (TDM) [6].

The TDM of TAC involves whole blood trough concentration (C_0_) measurements to adjust TAC dosage [6]. The golden standard in TAC C_0_ determination remains liquid chromatography-tandem mass spectrometry (LC-MS/MS) due to its higher sensitivity and precision compared to immunoenzymatic methods [6]. Despite almost 30 years of experience, the relationship between TAC C_0_ and clinical effects remains controversial [1,2,6]. This indicates insufficient accuracy of the whole blood C_0_ measurement as a predictor for rejection episodes [8,9,10] and adverse effects [11] related to TAC therapy.

TAC is also characterized by unpredictable pharmacokinetics, which result from several factors affecting its absorption and metabolism, and high affinity for binding with blood components [6,12]. Zahir et al. reported an extremely high TAC-bonding capacity of erythrocytes, which varied significantly between the 1st (average, 74.4%) and the 60th post-transplant day (average, 80.2%) among liver transplant recipients. In plasma, TAC was associated mostly with soluble proteins (61.2%), followed by HDL (28.1%), LDL (7.8%), and VLDL (1.4%) [11,13,14,15]. Only less than 0.5% and 3% of TAC is protein unbound in whole blood and in plasma, respectively [11,13,14].

It is believed that a novel approach to control and monitor TAC therapy may be useful. Zahir et al. reported that patients suffering TAC-related side effects presented significantly increased plasma TAC concentrations compared to those not experiencing toxicity (mean, standard deviation; 0.84 ± 0.19 ng/mL vs. 0.53 ± 0.19 ng/mL, respectively, *p* < 0.001), while whole blood TAC C_0_ was comparable (mean, standard deviation; 9.3 ± 2.2 ng/mL vs. 8.1 ± 1.8 ng/mL, respectively, *p* = 0.01) [11]. Measuring TAC directly in allogeneic organ tissue was recently proposed as it is reasonable to expect that local concentrations reflect drug effect most reliably than TAC whole blood C_0_ [6,16]. Considering that only protein-unbound drug particles are capable of crossing cellular membranes and achieving their active site, FK-binding protein [17], it is justified to assume that a free TAC concentration might be better associated with clinical outcomes. However, this requires thorough investigation. Free-drug concentration-adjusted TAC treatment is a potential target for improving and optimizing immunosuppressive treatment.

To date, it has been impossible to measure unbound TAC due to the very low drug concentration in pg/mL [18,19]. Only recently, an extremely sensitive LC-MS/MS method using a new approach to measure adducts in the positive ionization mode was developed, which allowed the measurement of TAC at a concentration range of 1–200 pg/mL [13,18,19]. Although the first descriptions of the method were published in 2016 [19] and 2018 [18], to date, the role of unbound TAC remains poorly investigated. In this study, we propose a novel, simple, and completely validated LC-MS/MS method for unbound TAC concentration measurements.

## 2. Materials and Methods

### 2.1. Chemicals and Patient Samples

TAC reference standard (99.1% purity) was purchased from Toronto Research Chemicals Inc. (Toronto, ON, Canada). Internal standard (IS), ascomycin (ASC, ≥90% purity) was from Sigma-Aldrich (St. Louis, MO, U.S.A.). Acetonitrile hypergrade, methanol, cyclohexane, and zinc sulfate heptahydrate were purchased from Merck KGaA (Darmstadt, Germany), and the ammonium fluoride assay was purchased from Sigma-Aldrich (St. Louis, MO, U.S.A.). Deionized water was obtained using Millipore SimPak^®^ 1, Simplicity 185 (Molsheim, France). Calibration curves were prepared using human plasma from healthy volunteers obtained from the Regional Blood Donation and Blood Treatment Center in Warsaw (Poland).

For this prospective study, blood samples were collected from 36 consecutive deceased kidney (*n* = 28) or liver (*n* = 8) transplant recipients operated on between August 2020 and March 2021 in a national transplant center. The postoperative immunosuppressive regimen was based on a once-daily or twice-daily TAC formula (Advagraf^®^ or Prograf^®^ Astellas Pharma, Warsaw, Poland). Patients requiring fluconazole or other agents that significantly influenced TAC metabolism were excluded from the study, as were non-adherent patients and multiorgan transplant recipients. EDTA-anticoagulated whole blood samples (12 mL) were collected on the 3rd, 5th, 7th, and 14th post-transplant days, resulting in a total of 140 samples (four of which were missed during follow-up). Detailed participant characteristics, inclusion and exclusion criteria, and study protocol descriptions are provided at ClinicalTrials.gov (NCT04657562).

Informed consent was obtained from all the patients. The study was approved by the Local Bioethical Committee (Medical University of Warsaw, KB/202/2018) and was performed in compliance with the Declaration of Helsinki, Council for International Organizations of Medical Sciences Guidelines, and the International Conference on Harmonization of Good Clinical Practice.

This study was funded by the National Science Center grant (2019/33/N/NZ7/01631).

### 2.2. Sample Preparation

Two milliliters of each whole blood sample were stored, and 10 mL was centrifuged at 3500 rpm at 37 °C for 10 min. After separation, 2 mL of plasma was injected into ultracentrifugation tubes (Ultra-Clear^®^, thinwall, 8 × 49 mm, Beckman, Brea, CA, U.S.A.) in thermoplastic adapters and inserted into a 12-place fixed-angle rotor (Type 70.1 Ti, Beckman Coulter, Indianapolis, IN, U.S.A.). Plasma samples were then centrifuged at 55,000 rpm at 37 °C for 5.5 h in an Optima^®^ L preparative ultracentrifuge (Beckman Coulter, Indianapolis, IN, U.S.A.). After preparation, approximately 800–1200 µL of clear plasma ultrafiltrate was placed into vials using low-binding tips and stored at −80 °C for 1–6 months.

For LC-MS/MS analysis, 10 µL of ASC (4.5 ng/mL) were added to 0.5 mL of ultrafiltrate sample to obtain a concentration of 90 pg/mL of IS. For protein precipitation, a mixture of aqueous zinc sulfate (0.1 mol/L) was used (acetonitrile:water:zinc sulfate/10:20:2 [*v*/*v*/*v*]). In the second step of sample purification, analyte extraction was performed with 1.5 mL of cyclohexane on a rotary mixer at 1500 rpm at room temperature for 20 min and centrifuged at 3500 rpm at 20 °C for 10 min. The sample was then evaporated under nitrogen in TURBO VAP BIOTAGE (50 °C water bath). The residue was reconstituted in 100 µL of methanol/water (1:1 [*v*/*v*]). The supernatant was then transferred to an autosampler vial and injected into the LC-MS/MS system.

### 2.3. LC–MS/MS Instrument Parameters

A Nexera X2 HPLC system (Shimadzu, Columbia, MO, U.S.A.) consisting of a binary pump (LC-30AD), degasser (DGU-20A5R), thermostatic column compartment (CTO-20AC), autosampler SIL-30AC (130 Pa), thermostat for the autosampler (CBM-20Alite), and an LCMS-8050 triple quadrupole MS (Shimadzu, Columbia, MO, U.S.A.) with a photodiode array detector (SPD-M30A) was used for sample analysis. It was also equipped with a 2-position/6-port rotary valve (FCV-20AH2), which is a stand-alone, high-pressure, flow-line selection control device. Chromatographic separation was performed using Hypurity™ C18 analytical column, 50 × 2.1 mm, 3 µm (Thermo Scientific, Vilnus, Lithuania), maintained at 60 °C, and guarded with HyPurity™ C18Drop-in Guards pk4, 10 × 2.1 mm, 3 µm precolumn (Thermo Scientific, Vilnus, Lithuania).

The mobile phase was a gradient of two solutions: (A) water with ammonium fluoride (2 mmol/L) and 0.05% formic acid and (B) methanol with ammonium fluoride (2 mmol/L) and 0.05% formic acid. The LC pump gradient program was set as follows:(1)A total of 90% of solution A and 10% of solution B directly after injection—minute 1.(2)A total of 5% of solution A and 95% of solution B—minutes 1 to 3.(3)A total of 90% of solution A and 10% of solution B—minutes 3 to 5.

Multiple reaction monitoring (MRM) of the immunosuppressants was performed using electrospray in positive ion mode. The ammonium adduct of each analyte [M + NH_4_]^+^ was monitored, using the mass transitions as follows: 821.5→768.4 *m*/*z* for TAC and 809.5→756.4 *m*/*z* for ASC.

The specific parameters were set at the following values: an electrospray voltage, 0.70 kV, the interface temperature, 300 °C; desolvation temperature, 526 °C; desolvation line temperature, 250 °C; and heat block temperature, 300 °C. The nebulizing gas, heating gas, and drying gas flow rates were 3 L/min, 10 L/min, and 10 L/min, respectively. Argon ≥ 99.99% (Multax S.C., Zielonki-Parcela, Poland) was used for the collision-induced dissociation at 270 kPa. The MS/MS conditions for each target were optimized using an automated MRM optimization procedure in LabSolutions (Shimadzu, Kyoto, Japan ). The MRM dwell time transition for TAC was 5–7 ms, with at least 12 measures per 3 s peak. The MS range for all the transitions was set to zero. Compound concentrations were calculated using a suitable IS. The second MRM transition of the analyzed compounds was not used in the calculations.

A brief summary of the proposed method is available in Figshare [20].

### 2.4. Method Validation

Method validation was performed in compliance with the European Medicines Agency (EMA) [21] and/or Food and Drug Administration (FDA) [22] guidelines, assessing the following parameters: selectivity, carry-over, lower limit of quantification (LLOQ), calibration curve, accuracy, precision, recovery, matrix effect, and stability effects. The parameters are consistent with both guidelines, otherwise the appropriate one is quoted. The nomenclature was based upon the EMA guideline.

Calibration standards (CSs) were prepared using the plasma of healthy volunteers (Section 2.1 and Section 2.2). The ultrafiltrate was spiked with the standard stock solution of TAC/methanol (0.1 µg/mL) to produce eight calibrators at the following concentrations: 0.1, 0.2, 1.0, 1.5, 3.0, 7.0, 10.0 and 20.0 pg/mL. An additional zero sample (containing only plasma ultrafiltrate and IS) and a blank sample (containing only plasma ultrafiltrate and no analytes) were prepared and analyzed. To assess inaccuracy and imprecision, quality control (QC) samples were prepared at the following concentrations: 0.5 pg/mL (lower QC—LQC), 1.5 pg/mL (medium QC—MQC), 10.0 pg/mL (higher QC1—HQC1), and 15.0 pg/mL (HQC2). All calibrators were then stored at −20 °C.

Selectivity was evaluated using six different samples of plasma ultrafiltrate without analytes, which were individually analyzed and evaluated for interference. Selectivity was considered adequate when the response of interfering components was lower than 20% and 5% of the LLOQ for the analyte and the IS, respectively.

The linearity of the method was assessed within the range of 0.1–20.0 pg/mL using 6 calibration curves [21]. Each curve consisted of eight CSs, a blank sample, and a zero sample. The LLOQ was tested using six different ultrafiltrate/TAC samples at a concentration of 0.1 pg/mL and was equal to the limit of detection (LOD). Specificity of CSs was tested using zero and blank samples. Two MRMs, for TAC and IS, were monitored to ensure that there was no interference in retention time. Six blank and zero samples were analyzed to detect possible interference with the analyte and IS peaks, respectively. Precision was determined at six concentration levels and expressed using the coefficient of variation. Accuracy was defined as the percentage difference between measured and nominal concentrations.

To assess the autosampler stability, each of the LQC (0.5 pg/mL) and HQC1 (10 pg/mL) were analyzed three times 0, 4, 8, and 24 h after preparation, when stored at 4 °C. Short-term stability was analyzed as described previously in a triple experiment [23,24]. Long-term stability was tested using LQC and HQC1 immediately after preparation and after the 2nd, the 3rd, and the 4th week of storage at 4 °C. The working solution stability (for TAC and IS) was analyzed immediately after preparation and then three times within 3-weeks observation when stored at 4 °C. Stability was considered acceptable when the difference in measured concentration was ±15% of the nominal value [22].

The analyte recovery (extraction efficiency) was tested by spiking equal amounts of TAC into aliquots of the plasma ultrafiltrate before and after extraction [22]. The experiment was performed using 0.5 pg/mL and 10 pg/mL TAC concentrations, each measured six times.

The carry-over effect was evaluated by analyzing the blank sample immediately after the measurement of the highest TAC CS (20 pg/mL) under the established chromatographic conditions [21]. Data were collected by examining seven sets of different analytical runs, including the IS.

The matrix effect (ME) was tested by producing six sets of QC samples and six sets of methanol-enriched samples with equal volumes of analyte and ISs added before or after the extraction step, as previously described [25,26,27], which was accepted by the FDA and EMA [21,22]. ME was evaluated using six different plasma samples for pre-extraction and post-extraction addition, by repeated measurements (*n* = 6) of reference solutions and IS in each experiment. The ME and process efficiency (PE) were calculated according to the method proposed by Taylor et al. [27].

### 2.5. Statistics

The data were analyzed using LabSolutions version 5.98 SP1 (Shimadzu, Columbia, MO, U.S.A.). All six calibration curves were analyzed individually by calculating a linear regression line using the least-squares method (1/x weighting), as it was best fitted for the concentration-detector response relationship.

## 3. Results

### 3.1. Method Development and Conditions

Unbound TAC determination in the plasma ultrafiltrate was performed using LC-MS/MS. The optimal analytical equipment was chosen for runtime, quality of peak shapes, and possibility of performing separation with the use of a simple mobile phase consisting of two solutions (ammonium fluoride and formic acid in water and methanol). Chromatographic parameters were experimentally established. The reaction temperature was set at 60 °C, with a flow rate of 0.75 mL/min. The total chromatographic runtime, including the column conditioning, was 5 min. The retention time was about ~1.5 min for each analyte (TAC and ASC). No interference was observed in the chromatograms.

Full-scan mass spectra examination of the two tacrolimus adducts, measured with positive ion ESI–MS Q1, was performed. The parent ions, product ions, collision energy, and radio-frequency lens were optimized in the laboratory. Chromatograms of the blank sample, LLOQ (0.1 pg/mL) sample, patient sample, and IS are presented in Figure 1.

### 3.2. Method Validation

No chromatographic interference was detected during selectivity analysis. The linearity of the method was evaluated for TAC concentrations within the range between 0.1 and 20 pg/mL using a set of six calibration curves. Each curve consisted of eight CSs at increasing concentrations, a blank sample, and zero sample. The calibration curves were calculated using the least-squares method (1/x), as this model showed no systematic deviations and the lowest summed absolute error for the tested QC concentrations. Calibration lines were characterized by an adequate coefficient of determination for IS: R^2^ = 0.9913 ± 0.013 (n = 6; mean ± standard deviation). Back-calculated TAC concentrations of CSs fell within an acceptable deviation ±15% of the nominal value (±20% for LLOQ). The LLOQ ensured the adequate accuracy and precision of this method for diagnostic purposes.

The accuracy and precision of this method were determined using the LLOQ, LQC, MQC, HQC1, and HQC2 samples. Within-run accuracies were 109.72%, 97.75%, 102.20%, 100.48%, and 97.33%, respectively. Between-run accuracies were 98.30%, 107.10%, 104.28%, 100.72%, and 100.75%, respectively. Accuracy and precision were also evaluated in the within- and between-run experiments for all four samples. All measured CSs and LLOQ fulfilled general analytical requirements of EMA (accuracy within 85–115%, imprecision less than 15%) [21] and also recent strict TDM recommendations (imprecision of at least ≤10%) [28]. Detailed results are presented in Table 1.

Autosampler stability was tested with the use of LQC and HQC1 samples after 24 h of storage at 4 °C. The measured concentrations were 97.96% and 103.16% of their initial values, respectively. The autosampler stability was within the EMA criteria.

Short-term stability was assessed with the use of triple experiments [23,24] at room temperature. No significant differences in QC samples concentration were observed. Additionally, TAC concentrations in all QCs remained stable in long-term storage at −20 °C. The LQC and HQC1 stability was 96.67% and 97.42% of the initial value after 4 weeks. Detailed stability results are summarized in Table 2.

TAC working solutions (0.01, 0.05, 0.15 and 1.0 ng/mL) were found to be very stable during the observation period lasting 3 weeks. The back-calculated concentrations were within ±15% of their initial values.

Carry-over effect for TAC was 0.1449% ± 0.0743% of the signal detected in the LLOQ sample. Carry-over for ASC was 0.0144% ± 0.0093%. No carry-over was observed in the blank samples analyzed immediately after HQC1.

The assay performance was not affected by intra-individual MEs. The outcomes of the experiment of pre-extraction/post-extraction addition with ion enhancement were: LQC 47.26% ± 5.13%/−46.26% ± 10.26% and HQC1 43.83% ± 3.76%/33.77% ± 21.63%, respectively. No MEs were observed during the entire chromatographic run. The detailed characteristics are presented in Table 3.

### 3.3. Patient Samples

This method was used to measure the unbound TAC concentration in plasma ultrafiltrate in 36 kidney and liver recipients (*n* = 28 and *n* = 8, respectively). The total number of measurements taken was 140. As a reference, whole blood TAC C_0_ was investigated. It ranged from 0.18 to 23.9 ng/mL (median: 4.84 ng/mL). Unbound tacrolimus in the patients’ plasma ultrafiltrate varied between 0.06 and 18.25 pg/mL (median: 0.98 pg/mL), which is presented in the dataset in Figshare [29], and was positively correlated with whole blood C_0_ (R^2^ = 0.44).

## 4. Discussion

In this study, we describe a new and highly sensitive method for determining unbound TAC. The proposed method was fully validated and had sufficient parameters for scientific research on the properties of unbound drug particles. It was also tested on samples from kidney or liver transplant recipients, showing adequate sensitivity, indicating that the method is also suitable for everyday TDM protocols.

The first attempts to develop similar methods have been reported recently [18,19]. In 2016, Stienestra et al. used Centrifree^®^ devices (Merck Millipore, Darmstadt, Germany) with a cellulose membrane to obtain a plasma ultrafiltrate. The proposed method was validated using calibrators and quality control samples prepared from newborn calf sera. Samples from the undefined group of five patients were measured showing concentrations in the range of 4.75–12.2 pg/mL [19]. Two years later, Bittersohl et al. proposed a complicated LS-MS/MS method using several expensive and not easily accessible tools, such as automatic sample clean-up. The unbound TAC concentrations covered the range of 1.1–7.9 pg/mL [18]. In both methods, linearity was determined using calibration points in the wide range of 1.00–200 pg/mL. Despite the possible significance of unbound TAC, to date, evidence remains limited [13]. Concentrations of the unbound drug are extremely low; therefore, several technical difficulties are encountered in developing quantitative methods. Thus, we share our experience in addressing the most complex issues.

The first step in the development of our method was to establish an appropriate matrix isolation procedure. Based on our experience with mycophenolic acid determination, we chose the Centrifree Micropartition System^®^ (Merck Millipore, Co., Cork, Ireland) used by Stienstra et al. [19]. However, despite the number of tests with different solvents (acetonitrile, methanol, and isopropanol), the measurements were not repeatable and we could not isolate the analyte properly. This was likely due to the non-specific binding of tacrolimus to the ultrafiltration membranes, as previously described [11,30]. Thus, we decided to ultracentrifuge using polyester terephthalate tubes (Ultra-Clear^®^, Beckman) and Optima^®^ L (Beckman Coulter, Indianapolis, IN, U.S.A.) as Bittersohl et al. [18]. After multiple testing, we established centrifugation conditions experimentally (55,000 rpm at 37 °C for 5.5 h), which allowed us to obtain between 800 and 1200 µL of clear ultrafiltrate.

Another step, the purification of the matrix, was performed using our experimental method, which was developed and optimized based on methods from previous publications [31,32,33]. Protein precipitation with a mixture of 0.1 mol/L aqueous zinc sulfate (acetonitrile:water:zinc sulfate/10:20:2 [*v*/*v*/*v*]) presented the best performance, but the level of purification was not sufficient. Subsequently, we performed a liquid–liquid extraction of the obtained solution, using organic solvents, such as tert-butyl ether, acetonitrile, hexane, isopropanol, and cyclohexane. Finally, adding 1.5 mL of cyclohexane, and then mixing and centrifuging it, yields the best results (Appendix A).

Because of the very low unbound TAC concentrations, we decided to use a nitrogen stream to concentrate the analyte, thus increasing the sensitivity of the method. After evaporation, the analyte was dissolved in (MeOH:H_2_O/1:1 [*v*/*v*]) and analyzed. The chromatogram shapes and peaks, recovery, and signal-to-noise ratio fell within the related EMA and FDA criteria [21,22].

Based on the evidence [18,19], we determined the linearity of the method using calibrators with concentrations in the range of 1.00–200 pg/mL. For verification, we analyzed 47 samples from renal and liver transplant recipients, but a substantial number of measurements did not fit the calibration curve. This was probably caused by the excessively wide range of the calibration curve compared to the extremely low unbound TAC concentrations in our patients (*n* = 140; range: 0.06–18.25 pg/mL).

Therefore, we needed to experimentally adjust the appropriate range of the calibration curve based on the unbound TAC concentrations obtained in the studied group. We decided to produce seven calibrators covering a range of 0.1–20 pg/mL, but the concentration of 0.1 pg/mL was below the sensitivity of the apparatus, according to the manufacturer’s instructions. To assess whether the MS/MS sensitivity was sufficient, the linearity of the method was compared with one-point calibration. Twelve separate solutions with a concentration of 0.1 pg/mL (*n* = 6) and 0.5 pg/mL (*n* = 6) were produced to perform a linearity experiment (Appendix A, Appendix A). This proves that our method is sufficiently sensitive, precise, and accurate for covering such low TAC concentrations. The method was verified with samples from patients, and all measurements fitted the curve and were noticeably different from the outcomes obtained using a wide range of concentrations of 1–200 pg/mL (Appendix A).

During the entire process, we used two ISs, ASC and deuterated and ^13^C-labeled tacrolimus (TAC^13^C,D_2_, ≥85% purity), purchased from Toronto Research Chemicals Inc. (Toronto, ON, Canada). Both models had similar validation parameters. However, the analysis of the TAC chromatograms showed that the measurements of the area under the concentration curve for TAC^13^C,D_2_ differed significantly. The reason was probably insufficient purity of IS, which is declared in the range of 85–90% and may affect the measurements at such low concentrations. Because of the lack of repeatability of the measurements with TAC^13^C, D_2_, we selected ASC as the IS in this study. A comparison of the chromatograms of TAC^13^C,D_2_, and ASC is presented in Appendix A.

Despite the current trend in the use of isotope-labeled ISs, ASC remains an exception [34]. To date, about 62% of laboratories use ASC as IS for TAC determination [35]. It may also be chosen due to economic aspects. Thus, we propose using ASC as the IS for unbound TAC measurement. However, it is necessary to adjust the isolation procedure as described previously [36].

In this study, we tried to follow the EMA and FDA guidelines. It has been easier since the last FDA guidelines were published. However, in light of the ongoing debate about the consistency and differences between the FDA and EMA guidelines, this study uses the EMA as a reference because of its well-established role in the scientific community [37]. Due to its popularity and simplicity, the nomenclature is also based on the EMA guideline.

We tested linearity according to the EMA guideline, as we did not provide the validation of ISs. However, due to its performance, we presented the results for ASC instead of isotope-labeled IS. The EMA was also used for the carry-over assessment for its more specific definition. On the other hand, recovery is poorly defined by the EMA, so we performed the test in line with the FDA guideline. The stability was also assessed according to more modern FDA references, assuming that the long-term stability study at −20 °C also covers lower temperatures.

## 5. Conclusions

We propose a new and highly sensitive method for determining unbound TAC. Issues related to the previously described methods have been addressed and solved. Ultracentrifugation remains the best tool for separating plasma ultrafiltrate containing unbound TAC from other blood compounds. We developed and optimized a two-step sample purification method that does not require complicated or expensive assays or equipment. The sensitivity of the apparatus was sufficient in a linearity range of 0.1–20 pg/mL, which allowed us to accurately measure the lowest unbound TAC concentrations in patients. Furthermore, ASC was the preferred IS over isotope-labeled TAC. This method is fast, easy, and requires the use of common solvents and standards. It can provide a basis for large-scale scientific research on the role of unbound TAC in transplant recipients, and because of its simplicity, it can be used in routine TDM protocols for post-transplant patient care.

## Figures and Tables

**Figure 1 pharmaceutics-14-00632-f001:**
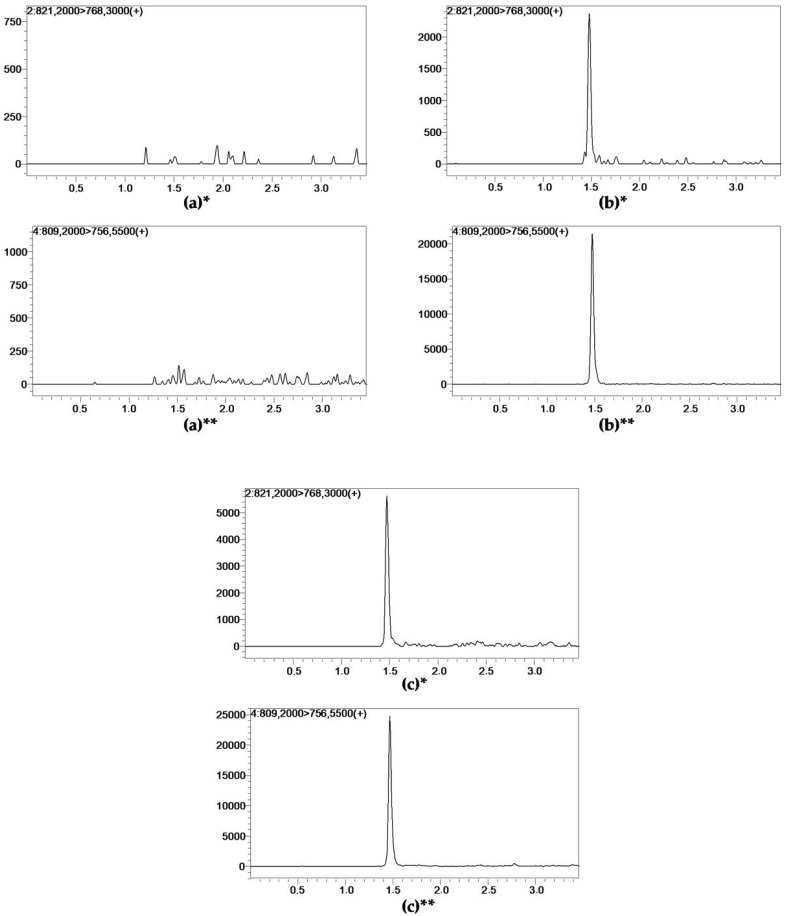
Representative LC-MS/MS chromatograms of (**a**) blank plasma ultrafiltrate; (**b**) tacrolimus at the lower limit of quantification (0.1 pg/mL); (**c**) unbound tacrolimus in a patient ultrafiltrate sample. * Tacrolimus reference standard (99.1% purity). ** Internal standard ascomycin.

**Table 1 pharmaceutics-14-00632-t001:** Within-run and between-run accuracy and precision (n = 6).

Sample	Concentration Declared [pg/mL]	Within-Run (*n* = 6)			Between-Run (*n* = 6)		
		Concentration [pg/mL]	Accuracy [%]	Imprecision [%]	Concentration [pg/mL]	Accuracy [%]	Imprecision [%]
LLOQ	0.1	0.12 ± 0.02	109.72	7.48	0.11 ± 0.03	98.30	13.74
LQC	0.5	0.49 ± 0.04	97.75	8.67	0.54 ± 0.06	107.10	10.67
MQC	1.5	1.54 ± 0.16	102.20	10.56	1.49 ± 0.31	104.28	9.07
HQC1	10	10.42 ± 0.27	100.48	2.67	10.07 ± 0.43	100.72	4.26
HQC2	15	14.60 ± 0.92	97.33	6.29	15.11 ± 0.55	100.75	3.64

Variables are expressed as mean ± SD or coefficient of variation (CV, %). HQC, high quality control; LLOQ, lower limit of quantification; LQC, low quality control; MQC, medium quality control.

**Table 2 pharmaceutics-14-00632-t002:** Summary of TAC stability in ultrafiltrate under different conditions (autosampler, short-term, and long-term stability).

Time	Low QC (0.5 pg/mL)		High QC1 (10 pg/mL)	
	Concentration Measured [pg/mL]	Stability [%]	Concentration Measured [pg/mL]	Stability [%]
Autosampler stability at 4 °C (*n* = 3)				
0 h	0.51 ± 0.03	100.00	10.14 ± 0.32	100.00
4 h	0.53 ± 0.03	104.56	9.80 ± 0.41	96.67
8 h	0.53 ± 0.08	104.38	10.04 ± 0.22	99.05
12 h	0.50 ± 0.06	97.96	10.26 ± 0.32	103.16
Short-term stability at room temperature (*n* = 3)				
0 h (standard procedure)	0.52 ± 0.01	100.00	10.03 ± 0.54	100.00
−4 h (before preparation)	0.50 ± 0.02	98.43	9.47 ± 0.36	94.42
+2 h (after preparation)	0.49 ± 0.02	95.09	10.08 ± 0.58	100.52
Long-term stability at −20 °C (*n* = 4)				
1 week	0.49 ± 0.03	100.00	10.16 ± 0.70	100.00
2 weeks	0.48 ± 0.10	99.32	10.23 ± 0.25	100.71
3 weeks	0.48 ± 0.07	98.54	10.26 ± 0.89	100.96
4 weeks	0.47 ± 0.05	96.67	9.90 ± 0.50	97.42

Variables are expressed as mean ± SD or coefficient of variation (CV, %). QC, quality control.

**Table 3 pharmaceutics-14-00632-t003:** Summary of the matrix effect and process efficiency.

Parameter	Low QC (0.5 pg/mL)			High QC1 (10 pg/mL)		
	TAC	IS ASC	F	TAC	IS ASC	F
ME [%] (n = 6)	−46.26 ± 10.26	−48.20 ± 9.98	−1.03 ± 0.06	−33.77 ± 21.63	−42.78 ± 19.36	16.42 ± 15.19
PE [%] (n = 6)	47.26 ± 5.13	49.86 ± 5.67	95.10 ± 7.97	43.83 ± 3.76	47.14 ± 4.56	95.42 ± 8.05

Variables are expressed as mean ± SD, F is calculated as TAC area/IS area. ASC, ascomycin; F, factor; IS, internal standard; ME, matrix effect; PE, process efficiency; QC, quality control; TAC, tacrolimus.

## Data Availability

The data presented in this study are openly available in FigShare at 10.6084/m9.figshare.19122482, reference number [20] and 10.6084/m9.figshare.19115243, reference number [29]. The data presented in this study are available in the Appendix A.

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
