# Peer review of "Development and Validation of the New Liquid Chromatography-Tandem Mass Spectrometry Method for the Determination of Unbound Tacrolimus in the Plasma Ultrafiltrate of Transplant Recipients"

_pharmaceutics, 2022, doi:10.3390/pharmaceutics14030632_

Round 1

Reviewer 1 Report

Dear Authors,

your article is well done and so interesting. However, needs minor revisions, before publication, because:

  • ® symbol in superscript in every words (lines 309 or 314 and others..)
  • Numbers of brute formula must to be at subscript (example, line 328)
  • In validation part, it is reported that you have done the validation following the FDA and EMA guidelines. In my opinion, is not clear  in what terms the method respects FDA or EMA guideline. Eg: FDA guidelines also provide the validation of the internal standard in linearity and variability terms...
  • There is need to divide the validations and specify the limits of both guidelines with the validation terms obtained.
  • Correct grammatical errors in table S2: "concentrations" and one space between words "whereas" and "only"
  •  
  •  

Reviewer 2 Report

Dear authors, I did enjoy going through the manuscript. It is well designed and well-executed, and well written. It covers all aspects of validating the new Liquid Chromatography-Tandem Mass Spectrometry Method. Thanks

Reviewer 3 Report

The manuscript "Development and validation of new liquid chromatography-tandem mass......." is a good scientific work of its kind. The authors have developed the research with rigor and scientific logic and have produced good results both on refence  and real samples, demonstrating the goodness of the proposed method. Unfortunately these are some inaccuracies that the authors would have to take seriously  in order to ensure that the work can be published. The first point is the following: the terms that must be used are LOD (limit of detection) and LOQ (limit of quantification) these is no lower limit of quantification; second point: the term within-run and between-run....is non correct but must be replaced with inter-day and intra-day which represents the conventional term. Once these corrections have been made, the work due to its good scientific content can be considered for pubblication on Pharmaceutics.
